# Pointer Graph Networks

**Petar Veličković, Lars Buesing, Matthew C. Overlan,**
**Razvan Pascanu, Oriol Vinyals and Charles Blundell**
DeepMind
{petarv,lbuesing,moverlan,razp,vinyals,cblundell}@google.com

## Abstract

Graph neural networks (GNNs) are typically applied to static graphs that are assumed to be known upfront. This static input structure is often informed purely by insight of the machine learning practitioner, and might not be optimal for the actual task the GNN is solving. In absence of reliable domain expertise, one might resort to inferring the latent graph structure, which is often difficult due to the vast search space of possible graphs. Here we introduce Pointer Graph Networks (PGNs) which augment sets or graphs with additional inferred edges for improved model generalisation ability. PGNs allow each node to dynamically *point* to another node, followed by message passing over these pointers. The sparsity of this adaptable graph structure makes learning tractable while still being sufficiently expressive to simulate complex algorithms. Critically, the pointing mechanism is directly supervised to model long-term sequences of operations on classical data structures, incorporating useful structural inductive biases from theoretical computer science. Qualitatively, we demonstrate that PGNs can learn parallelisable variants of pointer-based data structures, namely disjoint set unions and link/cut trees. PGNs generalise out-of-distribution to $5\times$ larger test inputs on dynamic graph connectivity tasks, outperforming unrestricted GNNs and Deep Sets.

## 1 Introduction

Graph neural networks (GNNs) have seen recent successes in applications such as quantum chemistry [14], social networks [32] and physics simulations [1, 23, 34]. For problems where a graph structure is known (or can be approximated), GNNs thrive. This places a burden upon the practitioner: which graph structure should be used? In many applications, particularly with few nodes, fully connected graphs work well, but on larger problems, sparsely connected graphs are required. As the complexity of the task imposed on the GNN increases and, separately, the number of nodes increases, not allowing the choice of graph structure to be data-driven limits the applicability of GNNs.

Classical algorithms [5] span computations that can be substantially more expressive than typical machine learning subroutines (e.g. matrix multiplications), making them hard to learn, and a good benchmark for GNNs [4, 8]. Prior work has explored learning primitive algorithms (e.g. arithmetic) by RNNs [57, 20, 45], neural approximations to NP-hard problems [50, 25], making GNNs learn (and transfer between) graph algorithms [47, 13], recently recovering a single neural core [55] capable of sorting, path-finding and binary addition. Here, we propose **Pointer Graph Networks** (PGNs), a framework that further *expands* the space of general-purpose algorithms that can be neurally executed.

Idealistically, one might imagine the graph structure underlying GNNs should be fully learnt from data, but the number of possible graphs grows *super-exponentially* in the number of nodes [39], making searching over this space a challenging (and interesting) problem. In addition, applying GNNs further necessitates the learning of the messages passed on top of the learnt structure, making it hard to disambiguate errors from having either the wrong structure or the wrong message. Several approaches have been proposed for searching over the graph structure [27, 16, 23, 21, 52, 9, 36, 19, 28]. Our

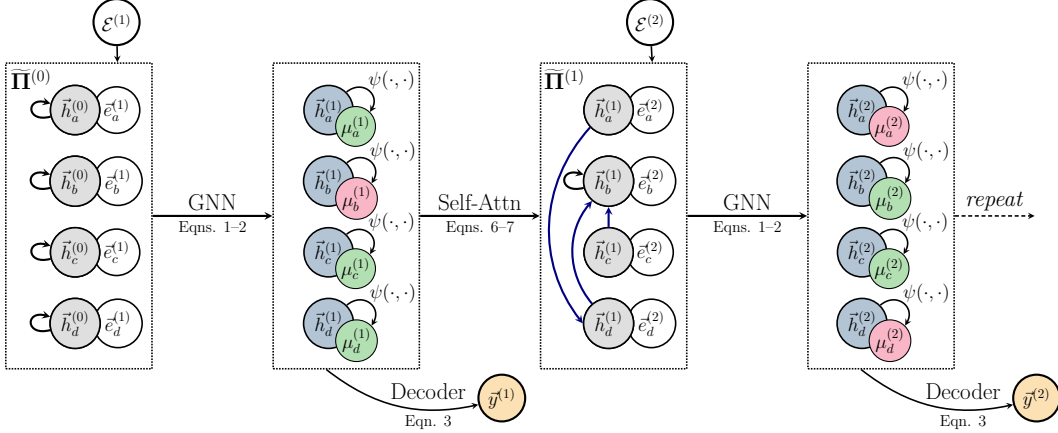

Figure 1: High-level overview of the pointer graph network (PGN) dataflow. Using descriptions of entity operations ($\vec{e}_i^{(t)}$), the PGN re-estimates latent features $\vec{h}_i^{(t)}$, masks $\mu_i^{(t)}$, and (asymmetric) pointers $\widetilde{\mathbf{\Pi}}^{(t)}$. The *symmetrised* pointers, $\mathbf{\Pi}^{(t)}$, are then used as edges for a GNN that computes next-step latents, $\vec{h}_i^{(t+1)}$, continuing the process. The latents may be used to provide answers, $\vec{y}^{(t)}$, to queries about the underlying data. We highlight masked out nodes in **red**, and modified pointers and latents in **blue**. See Appendix A for a higher-level visualisation, along with PGN's gradient flow.

PGNs take a hybrid approach, assuming that the practitioner may specify part of the graph structure, and then adaptively learn a linear number of *pointer* edges between nodes (as in [50] for RNNs). The pointers are optimised through *direct supervision* on classical data structures [5]. We empirically demonstrate that PGNs further increase GNN generalisation beyond those with static graph structures [12], without sacrificing computational cost or sparsity for this added flexibility in graph structure.

Unlike prior work on algorithm learning with GNNs [54, 47, 55, 6, 40], we consider algorithms that do not directly align to dynamic programming (making them inherently non-local [53]) and, crucially, the optimal known algorithms rely upon pointer-based data structures. The pointer connectivity of these structures dynamically changes as the algorithm executes. We learn algorithms that operate on two distinct data structures—disjoint set unions [11] and link/cut trees [38]. We show that baseline GNNs are unable to learn the complicated, data-driven manipulations that they perform and, through PGNs, show that extending GNNs with learnable dynamic pointer links enables such modelling.

Finally, the hallmark of having learnt an algorithm well, and the purpose of an algorithm in general, is that it may be applied to a wide range of input sizes. Thus by learning these algorithms we are able to demonstrate generalisation far beyond the size of training instance included in the training set.

Our PGN work presents *three* main contributions: we expand **neural algorithm execution** [54, 47, 55] to handle algorithms relying on complicated data structures; we provide a novel supervised method for sparse and efficient **latent graph inference**; and we demonstrate that our PGN model can deviate from the structure it is imitating, to produce **useful and parallelisable data structures**.

## 2 Problem setup and PGN architecture

**Problem setup**  We consider the following sequential supervised learning setting: Assume an underlying set of $n$ entities. Given are sequences of inputs $\mathcal{E}^{(1)}, \mathcal{E}^{(2)}, \ldots$ where $\mathcal{E}^{(t)} = (\vec{e}_1^{(t)}, \vec{e}_2^{(t)}, \ldots, \vec{e}_n^{(t)})$ for $t \geq 1$ is defined by a list of feature vectors $\vec{e}_i^{(t)} \in \mathbb{R}^m$ for every entity $i \in \{1, \ldots, n\}$. We will suggestively refer to $\vec{e}_i^{(t)}$ as an *operation* on entity $i$ at time $t$. The task consists now in predicting target outputs $\vec{y}^{(t)} \in \mathbb{R}^l$ based on operation sequence $\mathcal{E}^{(1)}, \ldots, \mathcal{E}^{(t)}$ up to $t$.

A canonical example for this setting is characterising *graphs with dynamic connectivity*, where inputs $\vec{e}_i^{(t)}$ indicate edges being added/removed at time $t$, and target outputs $\vec{y}^{(t)}$ are binary indicators of whether pairs of vertices are connected. We describe this problem in-depth in Section 3.

**Pointer Graph Networks**   As the above sequential prediction task is defined on the underlying, un-ordered set of entities, any generalising prediction model is required to be *invariant under permutations* of the entity set [49, 30, 56]. Furthermore, successfully predicting target $\vec{y}^{(t)}$ in general requires the prediction model to maintain a robust *data structure* to represent the history of operations for all entities throughout their lifetime. In the following we present our proposed prediction model, the Pointer Graph Network (PGN), that combines these desiderata in an efficient way.

At each step $t$, our PGN model computes *latent features* $\vec{h}_i^{(t)} \in \mathbb{R}^k$ for each entity $i$. Initially, $\vec{h}_i^{(0)} = \vec{0}$. Further, the PGN model determines dynamic *pointers*—one per entity and time step[1]—which may be summarised in a pointer adjacency matrix $\mathbf{\Pi}^{(t)} \in \mathbb{R}^{n \times n}$. Pointers correspond to undirected edges between two entities: indicating that one of them points to the other. $\mathbf{\Pi}^{(t)}$ is a binary symmetric matrix, indicating locations of pointers as 1-entries. Initially, we assume each node points to itself: $\mathbf{\Pi}^{(0)} = \mathbf{I}_n$. A summary of the coming discussion may be found in Figure 1.

The Pointer Graph Network follows closely the *encode-process-decode* [17] paradigm. The current operation is encoded together with the latents in each entity using an *encoder* network $f$:

$$\vec{z}_i^{(t)} = f\left(\vec{e}_i^{(t)}, \vec{h}_i^{(t-1)}\right) \tag{1}$$

after which the derived entity representations $\mathbf{Z}^{(t)} = (\vec{z}_1^{(t)}, \vec{z}_2^{(t)}, \dots, \vec{z}_n^{(t)})$ are given to a *processor network*, $P$, which takes into account the current pointer adjacency matrix as relational information:

$$\mathbf{H}^{(t)} = P\left(\mathbf{Z}^{(t)}, \mathbf{\Pi}^{(t-1)}\right) \tag{2}$$

yielding next-step latent features, $\mathbf{H}^{(t)} = (\vec{h}_1^{(t)}, \vec{h}_2^{(t)}, \dots, \vec{h}_n^{(t)})$; we discuss choices of $P$ below. These latents can be used to answer set-level queries using a *decoder* network $g$:

$$\vec{y}^{(t)} = g\left(\bigoplus_i \vec{z}_i^{(t)}, \bigoplus_i \vec{h}_i^{(t)}\right) \tag{3}$$

where $\bigoplus$ is any permutation-invariant *readout* aggregator, such as summation or maximisation.

Many efficient data structures only modify a small[2] subset of the entities at once [5]. We can incorporate this inductive bias into PGNs by masking their pointer modifications through a sparse *mask* $\mu_i^{(t)} \in \{0, 1\}$ for each node that is generated by a *masking* network $\psi$:

$$\mathbb{P}\left(\mu_i^{(t)} = 1\right) = \psi\left(\vec{z}_i^{(t)}, \vec{h}_i^{(t)}\right) \tag{4}$$

where the output activation function for $\psi$ is the logistic sigmoid function, enforcing the probabilistic interpretation. In practice, we threshold the output of $\psi$ as follows:

$$\mu_i^{(t)} = \mathbb{I}_{\psi\left(\vec{z}_i^{(t)}, \vec{h}_i^{(t)}\right) > 0.5} \tag{5}$$

The PGN now re-estimates the pointer adjacency matrix $\mathbf{\Pi}^{(t)}$ using $\vec{h}_i^{(t)}$. To do this, we leverage self-attention [46], computing all-pairs dot products between queries $\vec{q}_i^{(t)}$ and keys $\vec{k}_i^{(t)}$:

$$\vec{q}_i^{(t)} = \mathbf{W}_q \vec{h}_i^{(t)} \qquad \vec{k}_i^{(t)} = \mathbf{W}_k \vec{h}_i^{(t)} \qquad \alpha_{ij}^{(t)} = \text{softmax}_j\left(\left\langle \vec{q}_i^{(t)}, \vec{k}_j^{(t)} \right\rangle\right) \tag{6}$$

where $\mathbf{W}_q$ and $\mathbf{W}_k$ are learnable linear transformations, and $\langle \cdot, \cdot \rangle$ is the dot product operator. $\alpha_{ij}^{(t)}$ indicates the relevance of entity $j$ to entity $i$, and we derive the pointer for $i$ by choosing entity $j$ with the *maximal* $\alpha_{ij}$. To simplify the dataflow, we found it beneficial to *symmetrise* this matrix:

$$\widetilde{\mathbf{\Pi}}_{ij}^{(t)} = \mu_i^{(t)} \widetilde{\mathbf{\Pi}}_{ij}^{(t-1)} + \left(1 - \mu_i^{(t)}\right) \mathbb{I}_{j = \text{argmax}_k\left(\alpha_{ik}^{(t)}\right)} \qquad \mathbf{\Pi}_{ij}^{(t)} = \widetilde{\mathbf{\Pi}}_{ij}^{(t)} \vee \widetilde{\mathbf{\Pi}}_{ji}^{(t)} \tag{7}$$

where $\mathbb{I}$ is the indicator function, $\widetilde{\mathbf{\Pi}}^{(t)}$ denotes the pointers *before symmetrisation*, $\vee$ denotes logical disjunction between the two operands, and $1 - \mu_i^{(t)}$ corresponds to negating the mask. Nodes $i$ and $j$ will be linked together in $\mathbf{\Pi}^{(t)}$ (i.e., $\mathbf{\Pi}_{ij}^{(t)} = 1$) if $j$ is the most relevant to $i$, or vice-versa.

Unlike prior work which relied on the Gumbel trick [21, 23], we will provide direct supervision with respect to *ground-truth* pointers, $\hat{\mathbf{\Pi}}^{(t)}$, of a target data structure. Applying $\mu_i^{(t)}$ effectively *masks out* parts of the computation graph for Equation 6, yielding a *graph attention network*-style update [48].

Further, our data-driven *conditional masking* is reminiscent of neural execution engines (NEEs) [55]. Therein, masks were used to decide which inputs are participating in the computation at any step; here, instead, masks are used to determine which output state to overwrite, with all nodes participating in the computations at all times. This makes our model's hidden state end-to-end differentiable through all steps (see Appendix A), which was not the case for NEEs.

While Equation 6 involves computation of $O(n^2)$ coefficients, this constraint exists only at training time; at test time, computing entries of $\mathbf{\Pi}^{(t)}$ reduces to 1-NN queries in key/query space, which can be implemented storage-efficiently [31]. The attention mechanism may also be *sparsified*, as in [24].

**PGN components and optimisation** In our implementation, the encoder, decoder, masking and key/query networks are all linear layers of appropriate dimensionality—placing most of the computational burden on the *processor*, $P$, which explicitly leverages the computed pointer information.

In practice, $P$ is realised as a *graph neural network* (GNN), operating over the edges specified by $\mathbf{\Pi}^{(t-1)}$. If an additional *input graph* between entities is provided upfront, then its edges may be also included, or even serve as a completely separate "head" of GNN computation.

Echoing the results of prior work on algorithmic modelling with GNNs [47], we recovered strongest performance when using *message passing neural networks* (MPNNs) [14] for $P$, with elementwise maximisation aggregator. Hence, the computation of Equation 2 is realised as follows:

$$\vec{h}_i^{(t)} = U\left(\vec{z}_i^{(t)}, \max_{\mathbf{\Pi}_{ji}^{(t-1)}=1} M\left(\vec{z}_i^{(t)}, \vec{z}_j^{(t)}\right)\right) \qquad (8)$$

where $M$ and $U$ are linear layers producing *vector messages*. Accordingly, we found that elementwise-max was the best readout operation for $\bigoplus$ in Equation 3; while other aggregators (e.g. sum) performed comparably, maximisation had the least variance in the final result. This is in line with its *alignment* to the algorithmic task, as previously studied [33, 54]. We apply ReLU to the outputs of $M$ and $U$.

Besides the downstream query loss in $\vec{y}^{(t)}$ (Equation 3), PGNs optimise two additional losses, using *ground-truth* information from the data structure they are imitating: cross-entropy of the attentional coefficients $\alpha_{ij}^{(t)}$ (Equation 6) against the ground-truth pointers, $\hat{\mathbf{\Pi}}^{(t)}$, and binary cross-entropy of the masking network $\psi$ (Equation 4) against the ground-truth entities being modified at time $t$. This provides a mechanism for introducing domain knowledge in the learning process. At training time we readily apply *teacher forcing*, feeding ground-truth pointers and masks as input whenever appropriate. We allow gradients to flow from these auxiliary losses *back in time* through the latent states $\vec{h}_i^{(t)}$ and $\vec{z}_i^{(t)}$ (see Appendix A for a diagram of the backward pass of our model).

PGNs share similarities with and build on prior work on using latent $k$-NN graphs [9, 21, 52], primarily through addition of pointer-based losses against a ground-truth data structure and explicit entity masking—which will prove critical to generalising *out-of-distribution* in our experiments.

## 3    Task: Dynamic graph connectivity

We focus on instances of the *dynamic graph connectivity* setup to illustrate the benefits of PGNs. Even the simplest of structural detection tasks are known to be very challenging for GNNs [4], motivating dynamic connectivity as one example of a task where GNNs are unlikely to perform optimally.

Dynamic connectivity querying is an important subroutine in computer science, e.g. when computing *minimum spanning trees*—deciding if an edge can be included in the solution without inducing cycles [26], or *maximum flow*—detecting existence of paths from source to sink with available capacity [7].

Formally, we consider undirected and unweighted graphs of $n$ nodes, with evolving edge sets through time; we denote the graph at time $t$ by $G^{(t)} = (V, E^{(t)})$. Initially, we assume the graphs to be completely disconnected: $E^{(0)} = \emptyset$. At each step, an edge $(u, v)$ may be added to or removed from $E^{(t-1)}$, yielding $E^{(t)} = E^{(t-1)} \ominus \{(u, v)\}$, where $\ominus$ is the symmetric difference operator.

INIT$(u)$
1   $\hat{\pi}_u = u$
2   $r_u \sim \mathcal{U}(0,1)$

FIND$(u)$
1   **if** $\hat{\pi}_u \neq u$
2     $\hat{\pi}_u = \text{FIND}(\hat{\pi}_u)$
3   **return** $\hat{\pi}_u$

UNION$(u, v)$
1   $x = \text{FIND}(u)$
2   $y = \text{FIND}(v)$
3   **if** $x \neq y$
4     **if** $r_x < r_y$
5       $\hat{\pi}_x = y$
6     **else** $\hat{\pi}_y = x$

QUERY-UNION$(u, v)$
1   **if** $\text{FIND}(u) = \text{FIND}(v)$
2     **return** $0 \mathbin{/\!\!/} \hat{y}^{(t)} = 0$
3   **else** UNION$(u, v)$
4   **return** $1 \mathbin{/\!\!/} \hat{y}^{(t)} = 1$

Figure 2: Pseudocode of DSU operations; initialisation and `find(u)` (**Left**), `union(u, v)` (**Middle**) and `query-union(u, v)`, giving ground-truth values of $\hat{y}^{(t)}$ (**Right**). All manipulations of ground-truth pointers $\hat{\Pi}$ ($\hat{\pi}_u$ for node $u$) are in blue; the *path compression* heuristic is highlighted in green.

A *connectivity* query is then defined as follows: for a given pair of vertices $(u, v)$, does there exist a path between them in $G^{(t)}$? This yields binary ground-truth query answers $\hat{y}^{(t)}$ which we can supervise towards. Several classical data structures exist for answering variants of connectivity queries on dynamic graphs, on-line, in sub-linear time [11, 37, 38, 44] of which we will discuss two. All the derived inputs and outputs to be used for training PGNs are summarised in Appendix B.

**Incremental graph connectivity with *disjoint-set unions***   We initially consider *incremental* graph connectivity: edges can only be *added* to the graph. Knowing that edges can never get removed, it is sufficient to combine connected components through set union. Therefore, this problem only requires maintaining *disjoint sets*, supporting an efficient `union(u, v)` operation that performs a union of the sets containing $u$ and $v$. Querying connectivity then simply requires checking whether $u$ and $v$ are in the same set, requiring an efficient `find(u)` operation which will retrieve the set containing $u$.

To put emphasis on the data structure modelling, we consider a combined operation on $(u, v)$: first, query whether $u$ and $v$ are connected, then perform a union on them if they are not. Pseudocode for this `query-union(u, v)` operation is given in Figure 2 (Right). `query-union` is a key component of important algorithms, such as Kruskal's algorithm [26] for minimum spanning trees.—which was not modellable by prior work [47].

The tree-based *disjoint-set union* (DSU) data structure [11] is known to yield optimal complexity [10] for this task. DSU represents sets as *rooted trees*—each node, $u$, has a *parent pointer*, $\hat{\pi}_u$—and the set identifier will be its *root node*, $\rho_u$, which by convention points to itself ($\hat{\pi}_{\rho_u} = \rho_u$). Hence, `find(u)` reduces to recursively calling `find(pi[u])` until the root is reached—see Figure 2 (Left). Further, *path compression* [41] is applied: upon calling `find(u)`, all nodes on the path from $u$ to $\rho_u$ will point to $\rho_u$. This self-organisation substantially reduces future querying time along the path.

Calling `union(u, v)` reduces to finding the roots of $u$ and $v$'s sets, then making one of these roots point to the other. To avoid pointer ambiguity, we assign a random *priority*, $r_u \sim \mathcal{U}(0,1)$, to every node at initialisation time, then always preferring the node with higher priority as the new root—see Figure 2 (Middle). This approach of *randomised linking-by-index* [15] was recently shown to achieve time complexity of $O(\alpha(n))$ per operation in expectancy[3], which is optimal.

Casting to the framework of Section 2: at each step $t$, we call `query-union(u, v)`, specified by operation descriptions $\vec{e}_i^{(t)} = r_i \| \mathbb{I}_{i=u \vee i=v}$, containing the nodes' priorities, along with a binary feature indicating which nodes are $u$ and $v$. The corresponding output $\hat{y}^{(t)}$ indicates the return value of the `query-union(u, v)`. Finally, we provide supervision for the PGN's (asymmetric) pointers, $\widetilde{\Pi}^{(t)}$, by making them match the ground-truth DSU's pointers, $\hat{\pi}_i$ (i.e., $\hat{\Pi}_{ij}^{(t)} = 1$ iff $\hat{\pi}_i = j$ and $\hat{\Pi}_{ij}^{(t)} = 0$ otherwise.). Ground-truth mask values, $\hat{\mu}_i^{(t)}$, are set to 0 for only the paths from $u$ and $v$ to their respective roots—no other node's state is changed (i.e., $\hat{\mu}_i^{(t)} = 1$ for the remaining nodes).

Before moving on, consider how having access to these pointers *helps* the PGN answer queries, compared to a baseline without them: checking connectivity of $u$ and $v$ boils down to following their pointer links and checking if they meet, which drastically relieves learning pressure on its latent state.

QUERY-TOGGLE$(u, v)$

1   **if** $r_u < r_v$
2       SWAP$(u, v)$
3   EVERT$(u)$
4   **if** FIND-ROOT$(v) \neq u$
5       LINK$(u, v)$
6       **return** 0 // $\hat{y}^{(t)} = 0$
7   **else** CUT$(v)$
8   **return** 1 // $\hat{y}^{(t)} = 1$

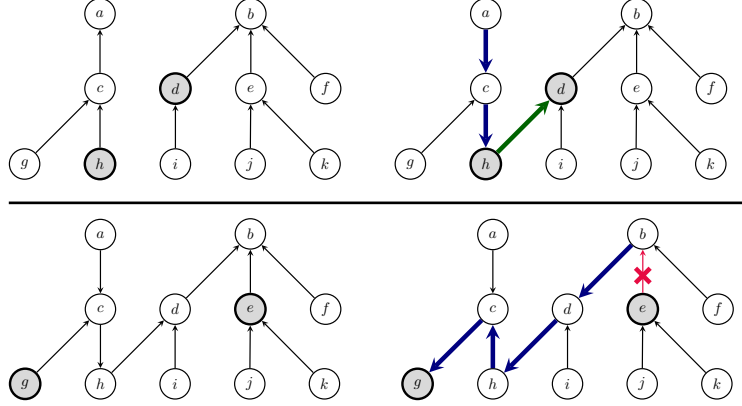

Figure 3: **Left:** Pseudocode of the `query-toggle(u, v)` operation, which will be handled by our models; **Right:** Effect of calling `query-toggle(h, d)` on a specific forest (**Top**), followed by calling `query-toggle(g, e)` (**Bottom**). Edges affected by `evert` (**blue**), `link` (**green**), and `cut` (**red**) are highlighted. **N.B.** this figure represents changes to the forest being modelled, and *not* the underlying LCT pointers; see Appendix C for more information on pointer manipulation.

**Fully dynamic tree connectivity with *link/cut trees***    We move on to *fully dynamic* connectivity—edges may now be removed, and hence set unions are insufficient to model all possible connected component configurations. The problem of fully dynamic *tree* connectivity—with the restriction that $E^{(t)}$ is acyclic—is solvable in amortised $O(\log n)$ time by *link/cut trees* (LCTs) [38], elegant data structures that maintain forests of *rooted trees*, requiring only one pointer per node.

The operations supported by LCTs are: `find-root(u)` retrieves the root of node $u$; `link(u, v)` links nodes $u$ and $v$, with the precondition that $u$ is the root of its own tree; `cut(v)` removes the edge from $v$ to its parent; `evert(u)` re-roots $u$'s tree, such that $u$ becomes the new root.

LCTs also support efficient *path-aggregate* queries on the (unique) path from $u$ to $v$, which is very useful for reasoning on dynamic trees. Originally, this speeded up bottleneck computations in network flow algorithms [7]. Nowadays, the LCT has found usage across online versions of many classical graph algorithms, such as *minimum spanning forests* and *shortest paths* [42]. Here, however, we focus only on checking connectivity of $u$ and $v$; hence `find-root(u)` will be sufficient for our queries.

Similarly to our DSU analysis, here we will compress updates and queries into one operation, `query-toggle(u, v)`, which our models will attempt to support. This operation first calls `evert(u)`, then checks if $u$ and $v$ are connected: if they are not, adding the edge between them wouldn't introduce cycles (and $u$ is now the root of its tree), so `link(u, v)` is performed. Otherwise, `cut(v)` is performed—it is guaranteed to succeed, as $v$ is not going to be the root node. Pseudocode of `query-toggle(u, v)`, along with visualising the effects of running it, is provided in Figure 3.

We encode each `query-toggle(u, v)` as $\vec{e}_i^{(t)} = r_i \| \mathbb{I}_{i=u \lor i=v}$. Random priorities, $r_i$, are again used; this time to determine whether $u$ or $v$ will be the node to call `evert` on, breaking ambiguity. As for DSU, we supervise the asymmetric pointers, $\widetilde{\mathbf{\Pi}}^{(t)}$, using the ground-truth LCT's pointers, $\hat{\pi}_i$ and ground-truth mask values, $\hat{\mu}_i^{(t)}$, are set to 0 only if $\hat{\pi}_i$ is modified in the operation at time $t$. Link/cut trees require elaborate bookkeeping; for brevity, we delegate further descriptions of their operations to Appendix C, and provide our C++ implementation of the LCT in the supplementary material.

## 4   Evaluation and results

**Experimental setup**    As in [47, 55], we evaluate *out-of-distribution generalisation*—training on operation sequences for small input sets ($n = 20$ entities with ops $= 30$ operations), then testing on up to $5\times$ larger inputs ($n = 50$, ops $= 75$ and $n = 100$, ops $= 150$). In line with [47], we generate 70 sequences for training, and 35 sequences across each test size category for testing. We generate operations $\vec{e}_i^{(t)}$ by sampling input node pairs $(u, v)$ uniformly at random at each step $t$; `query-union(u, v)` or `query-toggle(u, v)` is then called to generate ground-truths $\hat{y}^{(t)}, \hat{\mu}_i^{(t)}$

and $\hat{\mathbf{\Pi}}^{(t)}$. This is known to be a good test-bed for spanning many possible DSU/LCT configurations and benchmarking various data structures (see e.g. Section 3.5 in [42]).

All models compute $k = 32$ latent features in each layer, and are trained for $5,000$ epochs using Adam [22] with learning rate of $0.005$. We perform early stopping, retrieving the model which achieved the best query $F_1$ score on a validation set of 35 small sequences ($n = 20, \text{ops} = 30$). We attempted running the processor (Equation 8) for more than one layer between steps, and using a separate GNN for computing pointers—neither yielding significant improvements.

We evaluate the **PGN** model of Section 2 against three baseline variants, seeking to illustrate the benefits of its various graph inference components. We describe the baselines in turn.

**Deep Sets** [56] independently process individual entities, followed by an aggregation layer for resolving queries. This yields an only-self-pointer mechanism, $\mathbf{\Pi}^{(t)} = \mathbf{I}_n$ for all $t$, within our framework. Deep Sets are popular for set-based summary statistic tasks. Only the query loss is used.

**(Unrestricted) GNNs** [14, 35, 54] make no prior assumptions on node connectivity, yielding an all-ones adjacency matrix: $\mathbf{\Pi}_{ij}^{(t)} = 1$ for all $(t, i, j)$. Such models are a popular choice when relational structure is assumed but not known. Only the query loss is used.

**PGN** *without masks* **(PGN-NM)** remove the masking mechanism of Equations 4–7. This repeatedly overwrites all pointers, i.e. $\mu_i^{(t)} = 0$ for all $(i, t)$. PGN-NM is related to a directly-supervised variant of the prior art in learnable $k$-NN graphs [9, 21, 52]. PGN-NM has no masking loss in its training.

These models are *universal approximators* on permutation-invariant inputs [54, 56], meaning they are all able to model the DSU and LCT setup perfectly. However, unrestricted GNNs suffer from *oversmoothing* as graphs grow [58, 3, 51, 29], making it harder to perform robust *credit assignment* of relevant neighbours. Conversely, Deep Sets must process the entire operation history within their latent state, in a manner that is *decomposable* after the readout—which is known to be hard [54].

To assess the utility of the data structure learnt by the PGN mechanism, as well as its performance limits, we perform two tests with *fixed pointers*, supervised only on the query:

**PGN-Ptrs**: first, a PGN model is learnt on the training data. Then it is applied to derive and fix the pointers $\mathbf{\Pi}^{(t)}$ at all steps for all training/validation/test inputs. In the second phase, a new GNN over these inferred pointers is learnt and evaluated, solely on query answering.

**Oracle-Ptrs**: learn a query-answering GNN over the ground-truth pointers $\hat{\mathbf{\Pi}}^{(t)}$. Note that this setup is, especially for link/cut trees, made substantially easier than PGN: the model no longer needs to imitate the complex sequences of pointer rotations of LCTs.

**Results and discussion**   Our results, summarised in Table 1, clearly indicate outperformance and generalisation of our PGN model, especially on the larger-scale test sets. Competitive performance of PGN-Ptrs implies that the PGN models a robust data structure that GNNs can readily reuse. While the PGN-NM model is potent *in-distribution*, its performance rapidly decays once it is tasked to model larger sets of pointers at test time. Further, on the LCT task, baseline models often failed to make very meaningful advances at all—PGNs are capable of surpassing this limitation, with a result that even approaches ground-truth pointers with increasing input sizes.

We corroborate some of these results by evaluating pointer accuracy (w.r.t. ground truth) with the analysis in Table 2. Without masking, the PGNs fail to meaningfully model useful pointers on larger test sets, whereas the masked PGN consistently models the ground-truth to at least $50\%$ accuracy. Mask accuracies remain consistently high, implying that the inductive bias is well-respected.

Using the *max* readout in Equation 3 provides an opportunity for a qualitative analysis of the PGN's *credit assignment*. DSU and LCT focus on *paths* from the two nodes operated upon to their *roots* in the data structure, implying they are highly relevant to queries. As each global embedding dimension is pooled from *exactly* one node, in Appendix D we visualise how often these relevant nodes appear in the final embedding—revealing that the PGN's inductive bias *amplifies* their credit substantially.

**Rollout analysis of PGN pointers**   Tables 1–2 indicate a substantial *deviation* of PGNs from the ground-truth pointers, $\hat{\mathbf{\Pi}}^{(t)}$, while maintaining strong query performance. These learnt pointers

Table 1: $F_1$ scores on the dynamic graph connectivity tasks for all models considered, on five random seeds. All models are trained on $n = 20, \text{ops} = 30$ and tested on larger test sets.

| | Disjoint-set union | | | Link/cut tree | | |
|---|---|---|---|---|---|---|
| **Model** | $n = 20$ $\text{ops} = 30$ | $n = 50$ $\text{ops} = 75$ | $n = 100$ $\text{ops} = 150$ | $n = 20$ $\text{ops} = 30$ | $n = 50$ $\text{ops} = 75$ | $n = 100$ $\text{ops} = 150$ |
| GNN | $0.892_{\pm.007}$ | $0.851_{\pm.048}$ | $0.733_{\pm.114}$ | $0.558_{\pm.044}$ | $0.510_{\pm.079}$ | $0.401_{\pm.123}$ |
| Deep Sets | $0.870_{\pm.029}$ | $0.720_{\pm.132}$ | $0.547_{\pm.217}$ | $0.515_{\pm.080}$ | $0.488_{\pm.074}$ | $0.441_{\pm.068}$ |
| PGN-NM | $\mathbf{0.910}_{\pm.011}$ | $0.628_{\pm.071}$ | $0.499_{\pm.096}$ | $0.524_{\pm.063}$ | $0.367_{\pm.018}$ | $0.353_{\pm.029}$ |
| PGN | $0.895_{\pm.006}$ | $\mathbf{0.887}_{\pm.008}$ | $\mathbf{0.866}_{\pm.011}$ | $\mathbf{0.651}_{\pm.017}$ | $\mathbf{0.624}_{\pm.016}$ | $\mathbf{0.616}_{\pm.009}$ |
| PGN-Ptrs | $0.902_{\pm.010}$ | $0.902_{\pm.008}$ | $0.889_{\pm.007}$ | $0.630_{\pm.022}$ | $0.603_{\pm.036}$ | $0.546_{\pm.110}$ |
| Oracle-Ptrs | $0.944_{\pm.006}$ | $0.964_{\pm.007}$ | $0.968_{\pm.013}$ | $0.776_{\pm.011}$ | $0.744_{\pm.026}$ | $0.636_{\pm.065}$ |

Table 2: Pointer and mask accuracies of the PGN model w.r.t. ground-truth pointers.

| | Disjoint-set union | | | Link/cut tree | | |
|---|---|---|---|---|---|---|
| **Accuracy of** | $n = 20$ $\text{ops} = 30$ | $n = 50$ $\text{ops} = 75$ | $n = 100$ $\text{ops} = 150$ | $n = 20$ $\text{ops} = 30$ | $n = 50$ $\text{ops} = 75$ | $n = 100$ $\text{ops} = 150$ |
| Pointers (NM) | $\mathbf{80.3}_{\pm2.2\%}$ | $32.9_{\pm2.7\%}$ | $20.3_{\pm3.7\%}$ | $\mathbf{61.3}_{\pm5.1\%}$ | $17.8_{\pm3.3\%}$ | $8.4_{\pm2.1\%}$ |
| Pointers | $76.9_{\pm3.3\%}$ | $\mathbf{64.7}_{\pm6.6\%}$ | $\mathbf{55.0}_{\pm4.8\%}$ | $60.0_{\pm1.3\%}$ | $\mathbf{54.7}_{\pm1.9\%}$ | $\mathbf{53.2}_{\pm2.2\%}$ |
| Masks | $95.0_{\pm0.9\%}$ | $96.4_{\pm0.6\%}$ | $97.3_{\pm0.4\%}$ | $82.8_{\pm0.9\%}$ | $86.8_{\pm1.1\%}$ | $91.1_{\pm1.0\%}$ |

are still *meaningful*: given our 1-NN-like inductive bias, even minor discrepancies that result in modelling invalid data structures can have negative effects on the performance, if done uninformedly.

We observe the learnt PGN pointers on a pathological DSU example (Figure 4). Repeatedly calling `query-union(i, i+1)` with nodes ordered by priority yields a *linearised* DSU[4]. Such graphs (of large diameter) are difficult for message propagation with GNNs. During rollout, the PGN models a correct DSU at all times, but *halving* its depth—easing GNN usage and GPU parallelisability.

Effectively, the PGN learns to use the query supervision from $y^{(t)}$ to "nudge" its pointers in a direction more amenable to GNNs, discovering *parallelisable* data structures which may substantially deviate from the ground-truth $\hat{\mathbf{\Pi}}^{(t)}$. Note that this also explains the reduced performance gap of PGNs to Oracle-Ptrs on LCT; as LCTs cannot apply path-compression-like tricks, the ground-truth LCT pointer graphs are expected to be of substantially larger *diameters* as test set size increases.

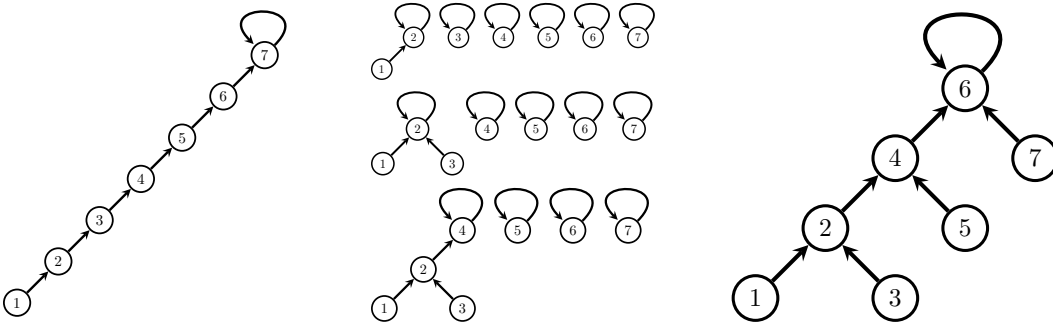

Figure 4: Visualisation of a PGN rollout on the DSU setup, for a pathological ground-truth case of repeated `union(i, i+1)` (**Left**). The first few pointers in $\mathbf{\Pi}^{(t)}$ are visualised (**Middle**) as well as the final state (**Right**)—the PGN produced a valid DSU at all times, but $2\times$ shallower than ground-truth.

Table 3: $F_1$ scores on the largest link/cut tree test set ($n = 100, \text{ops} = 150$) for four ablated models; the results on other datasets mirror these findings. GNN and PGN results reproduced from Table 1.

| GNN | SupGNN | DGM | PGN-MO | PGN-Asym | PGN |
|---|---|---|---|---|---|
| $0.401_{\pm.123}$ | $0.541_{\pm.059}$ | $0.524_{\pm.104}$ | $0.558_{\pm.022}$ | $0.561_{\pm.035}$ | $\mathbf{0.616}_{\pm.009}$ |

**Ablation studies** In Table 3, we provide results of several additional models, designed to probe additional hypotheses about the PGNs' contribution. These models are as follows:

**SupGNN** The contribution of our PGN model is two-fold: introducing *inductive biases* (such as pointers and masking) *and* usage of additional supervision (such as intermediate data structure rollouts). To verify that our gains do not arise from supervision alone, we evaluate **SupGNN**, the GNN model which features masking and pointer losses, but doesn't actually use this information. The model outperforms the GNN, while still being significantly behind our PGN results—implying our empirical gains are due to inductive biases as well.

**DGM** Our model's direct usage of data structure hints allows it to reason over highly relevant links. To illustrate the benefits of doing so, we also attempt training the pointers using the *differentiable graph module* (**DGM**) [21] loss function. DGM treats the model's downstream performance as a reward function for the chosen pointers. This allows it to outperform the baseline models, but not as substantially as PGN.

**PGN-MO** Conversely from our SupGNN experiment, our inductive biases can be strong enough to allow useful behaviour to emerge even in *limited supervision* scenarios. We were interested in how well the PGN would perform if we only had a sense of which data needs to be changed at each iteration—i.e. supervising on **m**asks **o**nly (**PGN-MO**) and letting the pointers adjust to nearest-neighbours (as done in [52]) without additional guidance. On average, our model outperforms all non-PGN models—demonstrating that even knowing only the mask information can be sufficient for PGNs to achieve state-of-the-art performance on out-of-distribution reasoning tasks.

**PGN-Asym** Our PGN model uses the *symmetrised* pointers $\mathbf{\Pi}$, where $i$ pointing to $j$ implies we will add both edges $i \to j$ and $j \to i$ for the GNN to use. This does not strictly align with the data structure, but we assumed that, empirically, it will allow the model to make mistakes more "gracefully", without disconnecting critical components of the graph. To this end, we provide results for **PGN-Asym**, where the pointer matrix remains asymmetric. We recover results that are significantly weaker than the PGN result, but still outperforming all other baselines on average. While this result demonstrates the empirical value of our approach to rectifying mistakes in the pointer mechanism, we acknowledge that better approaches are likely to exist—and we leave their study to future work.

Results are provided on the hardest LCT test set ($n = 100, \text{ops} = 150$); the results on the remaining test sets mirror these findings. We note that multiple GNN steps may be theoretically required to reconcile our work with teacher forcing the data structure. We found it sufficient to consider one-step in all of our experiments, but as tasks get more complex—especially when compression is no longer applicable—dynamic number of steps (e.g. function of dataset size) is likely to be appropriate.

Lastly, we scaled up the LCT test set to ($n = 200, \text{ops} = 300$) where the PGN ($0.636_{\pm.009}$) catches up to Oracle-Ptr ($0.619_{\pm.043}$). This illustrates not only the robustness of PGNs to larger test sets, but also provides a quantitative substantiation to our claim about ground-truth LCTs not having favourable diameters.

## 5 Conclusions

We presented **pointer graph networks** (PGNs), a method for simultaneously learning a latent pointer-based graph and using it to answer challenging algorithmic queries. Introducing step-wise structural supervision from classical data structures, we incorporated useful inductive biases from theoretical computer science, enabling outperformance of standard set-/graph-based models on two dynamic graph connectivity tasks, known to be challenging for GNNs. Out-of-distribution generalisation, as well as interpretable and parallelisable data structures, have been recovered by PGNs.

## Broader Impact

Our work evaluates the extent to which existing neural networks are potent reasoning systems, and the minimal ways (e.g. inductive biases / training regimes) to strengthen their reasoning capability. Hence our aim is not to outperform classical algorithms, but make their concepts accessible to neural networks. PGNs enable reasoning over edges not provided in the input, simplifying execution of any algorithm requiring a pointer-based data structure. PGNs can find direct practical usage if, e.g., they are pre-trained on known algorithms and then deployed on tasks which may require similar kinds of reasoning (with encoders/decoders "casting" the new problem into the PGN's latent space).

It is our opinion that this work does not have a specific immediate and predictable real-world application and hence no specific ethical risks associated. However, PGN offers a natural way to introduce domain knowledge (borrowed from data structures) into the learning of graph neural networks, which has the potential of improving their performance, particularly when dealing with large graphs. Graph neural networks have seen a lot of successes in modelling diverse real world problems, such as social networks, quantum chemistry, computational biomedicine, physics simulations and fake news detection. Therefore, indirectly, through improving GNNs, our work could impact these domains and carry over any ethical risks present within those works.

## Acknowledgments and Disclosure of Funding

We would like to thank the developers of JAX [2] and Haiku [18]. Further, we are very thankful to Danny Sleator for his invaluable advice on the theoretical aspects and practical applications of link/cut trees, and Abe Friesen, Daan Wierstra, Heiko Strathmann, Jess Hamrick and Kim Stachenfeld for reviewing the paper prior to submission.

This research was funded by DeepMind.

## Footnotes

[1]Chosen to match semantics of C/C++ pointers; a pointer of a particular type may have *exactly one* endpoint.

[2]Typically on the order of $O(\log n)$ elements.

[3] $\alpha$ is the *inverse Ackermann function*; essentially a constant for all sensible values of $n$. Making the priorities $r_u$ size-dependent recovers the optimal **amortised** time complexity of $O(\alpha(n))$ per operation [43].

[4]Note that this is not problematic for the ground-truth algorithm; it is constructed with a single-threaded CPU execution model, and any subsequent `find(i)` calls would result in path compression, amortising the cost.

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
