[Supplementary Material · NeurIPS_PGN_final_suppl.pdf]



Figure 5: Detailed view of the dataflow within the PGN model, highlighting inputs $\vec{e}_i^{(t)}$ (*outlined*), objects optimised against ground-truths (query answers $\vec{y}^{(t)}$, masks $\mu_i^{(t)}$ and pointers $\mathbf{\Pi}^{(t)}$) (*shaded*) and all intermediate latent states ($\vec{z}_i^{(t)}$ and $\vec{h}_i^{(t)}$). Solid lines indicate differentiable computation with gradient flow in **red**, while dotted lines indicate non-differentiable opeations (*teacher-forced* at training time). **N.B.** This computation graph should also include edges from $\vec{z}_i^{(t)}$ into the query answers, masks and pointers (as it gets concatenated with $\vec{h}_i^{(t)}$)—we omit these edges for clarity.

Table 4: Summary of operation descriptions and supervision signals on the data structures considered.

| Data structure and operation | Operation descriptions, $\vec{e}_i^{(t)}$ | Supervision signals |
|---|---|---|
| Disjoint-set union [11] query-union(u, v) | $r_i$: randomly sampled priority of node $i$, $\mathbb{I}_{i=u \vee i=v}$: Is node $i$ being operated on? | $\hat{y}^{(t)}$: are $u$ and $v$ in the same set?, $\hat{\mu}_i^{(t)}$: is node $i$ visited by find(u) or find(v)?, $\hat{\mathbf{\Pi}}_{ij}^{(t)}$: is $\hat{\pi}_i = j$ after executing? (*asymmetric* pointer) |
| Link/cut tree [38] query-toggle(u, v) | $r_i$: randomly sampled priority of node $i$, $\mathbb{I}_{i=u \vee i=v}$: Is node $i$ being operated on? | $\hat{y}^{(t)}$: are $u$ and $v$ connected?, $\hat{\mu}_i^{(t)}$: is node $i$ visited during query-toggle(u, v)?, $\hat{\mathbf{\Pi}}_{ij}^{(t)}$: is $\hat{\pi}_i = j$ after executing? (*asymmetric* pointer) |

# A   Pointer graph networks gradient computation

To provide a more high-level overview of the PGN model's dataflow across all relevant variables (and for realising its computational graph and differentiability), we provide the visualisation in Figure 5.

Most operations of the PGN are realised as standard neural network layers and are hence differentiable; the two exceptions are the thresholding operations that decide the final masks $\mu_i^{(t)}$ and pointers $\mathbf{\Pi}^{(t)}$, based on the soft coefficients computed by the masking network and the self-attention, respectively. This makes no difference to the training algorithm, as the masks and pointers are *teacher-forced*, and the soft coefficients are directly optimised against ground-truth values of $\hat{\mu}_i^{(t)}$ and $\hat{\mathbf{\Pi}}^{(t)}$.

Further, note that our setup allows a clear path to end-to-end backpropagation (through the latent vectors) at all steps, allowing the representation of $\vec{h}_i^{(t)}$ to be optimised with respect to *all* predictions made for steps $t' > t$ in the future.

# B   Summary of operation descriptions and supervision signals

To aid clarity, within Table 4, we provide an overview of all the operation descriptions and outputs (supervision signals) for the data structures considered here (disjoint-set unions and link/cut trees).

Note that the manipulation of ground-truth pointers ($\hat{\pi}_i$) is not discussed for LCTs in the main text for purposes of brevity; for more details, consult Appendix C.

# C   Link/cut tree operations

In this section, we provide a detailed overview of the link/cut tree (LCT) data structure [38], as well as the various operations it supports. This appendix is designed to be as self-contained as possible, and we provide the C++ implementation used to generate our dataset within the supplementary material.

Before covering the specifics of LCT operations, it is important to understand how it *represents* the forest it models; namely, in order to support efficient $O(\log n)$ operations and path queries, the pointers used by the LCT can differ significantly from the edges in the forest being modelled.

**Preferred path decomposition**    Many design choices in LCTs follow the principle of *"most-recent access"*: if a node was recently accessed, it is likely to get accessed again soon—hence we should keep it in a location that makes it easily accessible.

The first such design is *preferred path decomposition*: the modelled forest is partitioned into **preferred paths**, such that each node may have at most one *preferred child*: the child most-recently accessed during a node-to-root operation. As we will see soon, *any* LCT operation on a node $u$ will involve looking up the path to its respective root $\rho_u$—hence every LCT operation will be composed of several node-to-root operations.

One example of a preferred path decomposition is demonstrated in Figure 6 (Left). Note how each node may have *at most one* preferred child. When a node is not a preferred child, its parent edge is used to *jump between paths*, and is hence often called a **path-parent**.

**LCT pointers**    Each preferred path is represented by LCTs in a way that enables fast access—in a binary search tree (BST) keyed by depth. This implies that the nodes along the path will be stored in a binary tree (each node will potentially have a *left* and/or *right* child) which respects the following recursive invariant: for each node, all nodes in its left subtree will be *closer* to the root, and all nodes in its right subtree will be *further* from the root.

For now, it is sufficient to recall the invariant above—the specific implementation of binary search trees used in LCTs will be discussed towards the end of this section. It should be apparent that these trees should be *balanced*: for each node, its left and right subtree should be of (roughly) comparable sizes, recovering an optimal lookup complexity of $O(\log n)$, for a BST of $n$ nodes.

Each of the preferred-path BSTs will specify its own set of pointers. Additionally, we still need to include the *path-parents*, to allow recombining information across different preferred paths. While we could keep these links unchanged, it is in fact canonical to place the path-parent in the **root** node of the path's BST (**N.B.** this node may be different from the top-of-path node[5]!).

As we will notice, this will enable more elegant operation of the LCT, and further ensures that each LCT node will have *exactly one parent pointer* (either in-BST parent or path-parent, allowing for jumping between different path BSTs), which aligns perfectly with our PGN model assumptions.

The ground-truth pointers of LCTs, $\hat{\mathbf{\Pi}}^{(t)}$, are then recovered as all the parent pointers contained within these binary search trees, along with all the path-parents. Similarly, ground-truth masks, $\hat{\mu}_i^{(t)}$, will be the subset of LCT nodes whose pointers may change during the operation at time $t$. We illustrate how a preferred path decomposition can be represented with LCTs within Figure 6 (Right).

**LCT operations**    Now we are ready to cover the specifics of how individual LCT operations (find-root(u), link(u, v), cut(u) and evert(u)) are implemented.

Figure 6: **Left**: Rooted tree modelled by LCTs, with its four preferred paths indicated by solid lines. The most-recently accessed path is $f \rightarrow b \rightarrow a$. **Right**: One possible configuration of LCT pointers which models the tree. Each preferred path is stored in a binary search tree (BST) *keyed by depth* (colour-coded to match the LHS figure), and path-parents (dashed) emanate from the root node of each BST—hence their source node may changed (e.g. $d \twoheadrightarrow a$ is represented as $l \twoheadrightarrow a$).

All of these operations rely on an efficient operation which *exposes* the path from a node $u$ to its root, making it preferred—and making $u$ the root of the entire LCT (i.e. the root node of the top-most BST). We will denote this operation as `expose(u)`, and assume its implementation is provided to us for now. As we will see, *all* of the interesting LCT operations will necessarily start with calls to `expose(u)` for nodes we are targeting.

Before discussing each of the LCT operations, note one important invariant after running `expose(u)`: node $u$ is now the root node of the top-most BST (containing the nodes on the path from $u$ to $\rho_u$), and *it has no right children* in this BST—as it is the deepest node in this path.

As in the main document, we will highlight in **blue** all changes to the ground-truth LCT pointers $\hat{\pi}_u$, which will be considered as the union of ground-truth BST parents $\hat{p}_u$ and path-parents $\hat{pp}_u$. Note that each node $u$ will either have $\hat{p}_u$ or $\hat{pp}_u$; we will denote unused pointers with NIL. By convention, root nodes, $\rho$, of the entire LCT will point to themselves using a BST parent; i.e. $\hat{p}_\rho = \rho$, $\hat{pp}_\rho =$ NIL.

- `find-root(u)` can be implemented as follows: first execute `expose(u)`. This guarantees that $u$ is in the same BST as $\rho_u$, the root of the entire tree. Now, since the BST is keyed by depth and $\rho_u$ is the shallowest node in the BST's preferred path, we just follow *left children* while possible, starting from $u$: $\rho_u$ is the node at which this is no longer possible. We conclude with calling `expose` on $\rho_u$, to avoid pathological behaviour of repeatedly querying the root, accumulating excess complexity from following left children.

FIND-ROOT($u$)

1    EXPOSE($u$)
2    $\rho_u = u$
3    **while** left$_{\rho_u} \neq$ NIL // While currently considered node has left child
4        $\rho_u =$ left$_{\rho_u}$ // Follow left child links
5        EXPOSE($\rho_u$) // Re-expose to avoid pathological complexity
6    **return** $\rho_u$

- `link(u, v)` has the precondition that $u$ must be the root node of its respective tree (i.e. $u = \rho_u$), and $u$ and $v$ are not in the same tree. We start by running `expose(u)` and `expose(v)`. Attaching the edge $u \to v$ extends the preferred path from $v$ to its root, $\rho_v$, to incorporate $u$. Given that $u$ can have no left children in its BST (it is a root node, hence shallowest), this can be done simply by making $v$ a left child of $u$ (given $v$ is shallower than $u$ on the path $u \to v \to \ldots \to \rho_v$).

LINK($u, v$)

1    EXPOSE($u$)
2    EXPOSE($v$)
3    left$_u = v$ // $u$ cannot have left-children before this, as it is the root of its tree
4    $\hat{p}_v = u$ // $v$ cannot have parents before this, as it has been exposed

- `cut(u)`, as above, will initially execute `expose(u)`. As a result, $u$ will retain all the nodes that are deeper than it (through path-parents pointed to by $u$), and can just be cut off from all shallower nodes along the preferred path (contained in $u$'s left subtree, if it exists).

CUT($u$)

1    EXPOSE($u$)
2    **if** left$_u \neq$ NIL
3        $\hat{p}_{\text{left}_u} =$ left$_u$ // Cut off $u$ from left child, making it a root node of its component
4        left$_u =$ NIL

- `evert(u)`, as visualised in Figure 3, needs to isolate the path from $u$ to $\rho_u$, and *flip* the direction of all edges along it. The first part is already handled by calling `expose(u)`, while the second is implemented by recursively *flipping* left and right subtrees within the entire BST containing $u$ (this makes shallower nodes in the path become deeper, and vice-versa).

  This is implemented via *lazy propagation*: each node $u$ stores a *flip bit*, $\phi_u$ (initially set to 0). Calling `evert(u)` will toggle node $u$'s flip bit. Whenever we process node $u$, we further issue a call to a special operation, `release(u)`, which will perform any necessary flips of $u$'s left and right children, followed by propagating the flip bit onwards. Note that `release(u)` *does not* affect parent-pointers $\hat{\pi}_u$—but it may affect outcomes of future operations on them.

RELEASE($u$)

1    **if** $u \neq$ NIL **and** $\phi_u = 1$
2        SWAP(left$_u$, right$_u$) // Perform the swap of $u$'s left and right subtrees
3        **if** left$_u \neq$ NIL
4            $\phi_{\text{left}_u} = \phi_{\text{left}_u} \oplus 1$ // Propagate flip bit to left subtree
5        **if** right$_u \neq$ NIL
6            $\phi_{\text{right}_u} = \phi_{\text{right}_u} \oplus 1$ // Propagate flip bit to right subtree
7        $\phi_u = 0$

EVERT($u$)

1    EXPOSE($u$)
2    $\phi_u = \phi_u \oplus 1$ // Toggle $u$'s flip bit ($\oplus$ is binary exclusive OR)
3    RELEASE($u$) // Perform lazy propagation of flip bit from $u$

**Implementing `expose(u)`**    It only remains to provide an implementation for `expose(u)`, in order to specify the LCT operations fully.

Figure 7: A schematic of a *zig-zag* rotation: first, node $u$ is rotated around node $v$; then, node $u$ is rotated around node $w$, bringing it two levels higher in the BST without breaking invariants.

LCTs use *splay trees* as the particular binary search tree implementation to represent each preferred path. These trees are also designed with "most-recent access" in mind: nodes recently accessed in a splay tree are likely to get accessed again, therefore any accessed node is turned into the *root node* of the splay tree, using the `splay(u)` operation. The manner in which `splay(u)` realises its effect is, in turn, via a sequence of complex *tree rotations*; such that `rotate(u)` will perform a rotation that brings $u$ one level higher in the tree.

We describe these three operations in a *bottom-up* fashion: first, the lowest-level `rotate(u)`, which merely requires carefully updating all the pointer information. Depending on whether $u$ is its parent's left or right child, a *zig* or *zag* rotation is performed—they are entirely symmetrical. Refer to Figure 7 for an example of a *zig* rotation followed by a *zag* rotation (often called *zig-zag* for short).

ROTATE($u$)

```
 1   v = p̂_u
 2   w = p̂_v
 3   if left_v = u // Zig rotation
 4        left_v = right_u
 5        if right_u ≠ NIL
 6             p̂_{right_u} = v
 7        right_u = v
 8   else // Zag rotation
 9        right_v = left_u
10        if left_u ≠ NIL
11             p̂_{left_u} = v
12        left_u = v
13   p̂p_u = p̂p_v
14   p̂_v = u
15   p̂p_v = NIL
16   if w ≠ NIL // Adjust grandparent
17        if left_w = v
18             left_w = u
19        else
20             right_w = u
21   p̂_u = w
```

Armed with the rotation primitive, we may define `splay(u)` as a repeated application of *zig*, *zig-zig* and *zig-zag* rotations, until node $u$ becomes the root of its BST[6]. We also repeatedly perform *lazy propagation* by calling `release(u)` on any encountered nodes.

SPLAY($u$)

1    **while** $\hat{p}_u \neq$ NIL // Repeat while $u$ is not BST root
2        $v = \hat{p}_u$
3        $w = \hat{p}_v$
4        RELEASE($w$) // Lazy propagation
5        RELEASE($v$)
6        RELEASE($u$)
7        **if** $w =$ NIL // *zig* or *zag* rotation
8           ROTATE($u$)
9        **if** $(\text{left}_w = v) = (\text{left}_v = u)$ // *zig-zig* or *zag-zag* rotation
10          ROTATE($v$)
11          ROTATE($u$)
12       **else** // *zig-zag* or *zag-zig* rotation
13          ROTATE($u$)
14          ROTATE($u$)
15    RELEASE($u$) // In case $u$ was root node already

Finally, we may define `expose(u)` as repeatedly interchanging calls to `splay(u)` (which will render $u$ the root of its preferred-path BST) and appropriately following path-parents, $\hat{pp}_u$, to fuse $u$ with the BST above. This concludes the description of the LCT operations.

EXPOSE($u$)

1    **do**
2        SPLAY($u$) // Make $u$ root of its BST
3        **if** $\text{right}_u \neq$ NIL // Any deeper nodes than $u$ along preferred path are no longer preferred
4           $\hat{p}_{\text{right}_u} =$ NIL // They get cut off into their own BST
5           $\hat{pp}_{\text{right}_u} = u$ // This generates a new path-parent into $u$
6           $\text{right}_u =$ NIL
7        $w = \hat{pp}_u$ // $u$ is either LCT root or it has gained a path-parent by splaying
8        **if** $w \neq$ NIL // Attach $u$ into $w$'s BST
9           SPLAY($w$) // First, splay $w$ to simplify operation
10          **if** $\text{right}_w \neq$ NIL // Any deeper nodes than $w$ are no longer preferred; detach them
11             $\hat{p}_{\text{right}_w} =$ NIL
12             $\hat{pp}_{\text{right}_w} = w$
13          $\text{right}_w = u$ // Convert $u$'s path-parent into a parent
14          $\hat{p}_u = w$
15          $\hat{pp}_u =$ NIL
16    **while** $\hat{p}_u \neq u$ // Repeat until $u$ is root of its LCT

It is worth reflecting on the overall complexity of individual LCT operations, taking into account the fact they're propped up on `expose(u)`, which itself requires reasoning about *tree rotations*, followed by appropriately leveraging preferred path decompositions. This makes the LCT modelling task substantially more challenging than DSU.

**Remarks on computational complexity and applications**    As can be seen throughout the analysis, the computational complexity of all LCT operations can be reduced to the computational complexity of calling `expose(u)`—adding only a constant overhead otherwise. `splay(u)` has a known amortised complexity of $O(\log n)$, for $n$ nodes in the BST; it seems that the ultimate complexity of exposing is this multiplied by the worst-case number of different preferred-paths encountered.

However, detailed complexity analysis can show that splay trees combined with preferred path decomposition yield an amortised time complexity of *exactly* $O(\log n)$ for *all* link/cut tree operations. The storage complexity is highly efficient, requiring $O(1)$ additional bookkeeping per node.

Finally, we remark on the utility of LCTs for performing *path aggregate* queries. When calling `expose(u)`, all nodes from $u$ to the root $\rho_u$ become exposed in the same BST, simplifying computations of important path aggregates (such as bottlenecks, lowest-common ancestors, etc). This can be augmented into an arbitrary `path(u, v)` operation by first calling `evert(u)` followed by `expose(v)`—this will expose only the nodes along the *unique* path from $u$ to $v$ within the same BST.

Figure 8: Credit assignment study results for the DSU setup, for the baseline GNN (**Top**) and the PGN (**Bottom**), arranged left-to-right by test graph size. PGNs learn to put larger emphasis on both the two nodes being operated on (**blue**) and the nodes on their respective paths-to-roots (**green**).

Figure 9: Credit assignment study results for the LCT setup, following same convention as Figure 8.

## D   Credit assignment analysis

Firstly, recall how our decoder network, $g$, is applied to the latent state $(\vec{z}_i^{(t)}, \vec{h}_i^{(t)})$, in order to derive predicted query answers, $\vec{y}_i^{(t)}$ (Equation 3). Knowing that the *elementwise maximisation* aggregator performed the best as aggregation function, we can rewrite Equation 3 as follows:

$$\vec{y}^{(t)} = g\left(\max_i \vec{z}_i^{(t)}, \max_i \vec{h}_i^{(t)}\right) \tag{9}$$

This form of *max-pooling* readout has a unique feature: each dimension of the input vectors to $g$ will be contributed to by *exactly* one node (the one which optimises the corresponding dimension in $\vec{z}_i^{(t)}$ or $\vec{h}_i^{(t)}$). This provides us with opportunity to perform a *credit assignment* study: we can verify how often every node has propagated its features into this vector—and hence, obtain a direct estimate of how "useful" this node is for the decision making by any of our considered models.

We know from the direct analysis of disjoint-set union (Section 3) and link/cut tree (Appendix C) operations that only a subset of the nodes are directly involved in decision-making for dynamic

connectivity. These are exactly the nodes along the *paths* from $u$ and $v$, the two nodes being operated on, to their respective *roots* in the data structure. Equivalently, these nodes directly correspond to the nodes tagged by ground-truth masks (nodes $i$ for which $\hat{\mu}_i^{(t)} = 0$).

With the above hindsight, we compare a trained baseline GNN model against a PGN model, in terms of how much *credit* is assigned to these "important" nodes, throughout the rollout. The results of this study are visualised in Figures 8 (for DSU) and 9 (for LCT), visualising separately the credit assigned to the two nodes being operated on (**blue**) and the remaining nodes along their paths-to-roots (**green**).

From these plots, we can make several direct observations:

- In all settings, the PGN *amplifies* the overall credit assigned to these relevant nodes.
- On the DSU setup, the baseline GNN is likely suffering from *oversmoothing* effects: at larger test set sizes, it seems to hardly distinguish the paths-to-root (which are often very short due to path compression) from the remainder of the neighbourhoods. The PGN explicitly encodes the inductive bias of the structure, and hence more explicitly models such paths.
- As ground-truth LCT pointers are not amenable to path compression, paths-to-root may more significantly grow in lengthwith graph size increase. Hence at this point the oversmoothing effect is less pronounced for baselines; but in this case, LCT operations are highly centered on the node being operated on. The PGN learns to provide additional emphasis to the nodes operated on, $u$ and $v$.

In all cases, it appears that through a careful and targeted constructed graph, the PGN is able to significantly overcome the oversmoothing issues with fully-connected GNNs, providing further encouragement for applying PGNs in problem settings where strong credit assignment is required, one example of which are *search* problems.

## Footnotes

[5]The top-of-path node is always the *minimum* node of the BST, obtained by recursively following left-children, starting from the root node, while possible.

[6]Note: this exact sequence of operations is required to achieve optimal amortised complexity.