[Reviews · NeurIPS 2020]

Review 1

Summary and Contributions: This paper introduces an algorithm based on Graph Neural Networks to learn with strong supervision from two complex classical data structures the task of dynamic connectivity. This had not been explored before. It shows that the algorithm performs better than other neural baselines (unstructured GNNs and Deep Sets).

Strengths: The task and domain tackled by the authors is important as current neural models are not very capable of performing algorithmic tasks. The model is somewhat complicated and the authors do a good job at writing the paper, explaining the model and making references to the literature. The many modules in the model make it a complicated paper/model. Making so many components work properly is a strength in itself. The tasks tried mark an improvement over existing previous work on neural algorithmic execution. The overall framework of training and the model are novel.

Weaknesses: The results of the paper might be somewhat weak. They claim they are making three contributions. 1. Expand the Neural algorithm execution. While this works explore new tasks in a novel way, the numbers are not too impressive, specially since we have classical algorithms which can perform this task. How would does performance decrease with even longer and harder out of distribution tasks? Is there a point where the neural engine might be better than the classical counterpart? The baselines also do not receive the same type of supervision so it could be expected that they would perform weaker. If the contribution is the idea of using supervision itself (as opposed to the particular algorithm), then this perhaps could be stated more clearly. But if the framing is about the contribution and extension to new algorithmic tasks, then the results are not particularly impressive and the comparison to baselines seems not entirely fair as they receive less supervision. 2. Learn latent graph. This is somewhat contained in contribution 1 as it is necessary to enable it. As a contribution on its own, a deeper empirical comparison in previously explored tasks with other algorithms would be good. 3. Can deviate from the structures it is imitating. The authors claim this but do not analyze it very systematically. In particular they say “this also explains the reduced performance gap of PGNs from the ground-truth to Oracle-Ptrs on LCT; as LCTs cannot apply path-compression-like tricks, the ground-truth LCT pointer graphs are expected to be of substantially larger diameters as test set size increases.” This is an interesting an important claim, but a bit more systematic quantitative analysis would be good if it is intended to be presented as an important contribution of the paper.

Correctness: The paper seems correct.

Clarity: Yes, it is well written, being a somewhat complicated paper I think the authors do a good job.

Relation to Prior Work: It does a very good job at explaining how it relates to previous literature through out the paper.

Reproducibility: Yes

Additional Feedback: A discussion of why making neural algorithms that replicate this tasks when there are already existing classical structures would be desirable. While there are potentially many benefits of a neural learning algorithm, some discussion clarifying the authors perspective would be desirable and useful. ----------------------------------------------------- Post Rebuttal: I would like to reafirm my score. While there are potentially good arguments in favor of neural algorithms, these are general and I don’t think should automatically transfer to any paper converting something classical into neural without a particularly interesting innovation. And would not like to see papers turning any algorithm into neural. It happened before for example with Neural Turing Machines, Neural Program Interpreters, Neural GPU’s, etc. And this line of research (while highly cited) does not seem to be particularly insightful or interesting now, specially not all the papers that came after the initial one (which did present a promising and interesting direction). Having said that, I still think this paper is solid and sufficiently innovative.


Review 2

Summary and Contributions: This paper proposes pointer graph networks (PGN) for inferring the dynamic latent structures underlying sequential data. Several model designs were made to ensure the learned structures are meaningful and sparse. The major application task of PGN introduced in this paper is dynamic graph connectivity.

Strengths: 1. the problem of inferring dynamic structures of data where graphs are absent is important for modeling the correlations among samples 2. message passing over the inferred graphs 3. self-attention based pointer selection and gated/masked transformation of learned graphs

Weaknesses: 1. the practical usefulness of the proposed method is obscure 2. the assumption of ground truth pointers and ground truth masks may limit the generalization of the method in wide applications 3. the simplified symmetric assumption of the pointer graph may lead to information loss

Correctness: The method derivation is solid. Method and empirical claims are correct.

Clarity: Yes

Relation to Prior Work: Yes

Reproducibility: Yes

Additional Feedback: The paper provides a good algorithm to infer data structures, with sufficient supervision. The major question to this work is about PGN's practical meaning. Given the problem description, and the assumption on ground truth pointers and ground truth masks, I feel it is hard to link the problem to physical applications. The major task described on disjoint set unions and link/cut trees is somewhat narrow, not very clear on the impacts of these tasks and how to generalize them to solve real-world problems. Some other questions are as below. 1. Is it beneficial to consider directed pointer graph, i.e., asymmetric matrix, with real-valued weights? 2. Due to the similarity between the proposed method and the method in [19], which also infers structures underlying sequential data. Is it possible to compare the proposed method with it on some datasets? EDIT AFTER READING AUTHOR FEEDBACK Thanks to the authors for providing some details in the rebuttal. From the paper and the feedback, I agree with other reviewers that more discussions are still expected on practical applications, novelty compared with [42] [50], and clarifications on its value. Therefore, I'd like to keep my original score.


Review 3

Summary and Contributions: The paper presents a novel GNN architecture - Pointer Graph Network (PGN). In most existing GNNs, the structure is static - edges are given as part of the input and remain constant. In contrast, the proposed architecture also predicts the adjacency between the nodes in the next propagation step. This allows the new PGN architecture to simulate the execution of operations on data structures like union-find (AKA disjoint sets), and at test time, even generalize to larger graphs than were observed at training time. Another (simpler?) way to think about it is as a sequence of GNN layers, intertwined with fully connected layers (layers where every node is connected to every other node, a "complete graph"): GNN->FC->GNN->GC-> ... The fully-connected layers classify whether node u should point to node v in the next GNN layer, for every pair of nodes (u,v). The entire network classifies whether two given (u,v) nodes are in the same set (in the first task) or whether the given (u,v) nodes are connected (in the second task). Additionally, the GNN layer also performs a binary classification for every node pair - of whether or not their current adjacency should be modified (overriding the adjacency prediction of the fully-connected layer). There are 3 classification losses that are applied at every time step, supervised by the execution of the oracle algorithm.

Strengths: * The shift from static graphs to dynamic graphs, the idea of letting the GNN predicts its own adjacency, is very interesting (although it was partially introduced in prior work, see below). * The evaluation is performed on two tasks (disjoint sets, and link/cut-tree), and includes two baselines (GNN, Deep Sets) and 3 ablations (no-masks, two-steps, and oracle-pointers). The generalization to 5x larger graphs shows evidence that the model has actually learned the "rules" that perform the algorithmic operations. * The complexity of PGN is O(n^2), while most GNNs work in O(|E|) (the number of edges). This may sound like a weakness, but to me, this is perfectly OK, as long as the authors discussed it (L103). Further, the number of nodes in the problems that this paper addresses is around n=100, which can afford the quadratic computation.

Weaknesses: * Much of the supposedly novelty in this paper is shared with prior work (cited as [42] and [50] in this paper), as detailed in the "Relations to prior work" section below. * I find the main weakness of this paper to be its applicability. Since the PGN model requires heavy and direct supervision at every time step - at best, it just learns to mimic the original algorithm (the data structure's operation). Typically, learning models are useful in tasks where sound-and-complete algorithms do not exist. In the case of this paper, what benefit do we have in having a neural network that simulates what the known algorithm is already doing perfectly? If training the model required only the final supervision (only "whether or not u and v are connected", without the intermediate supervision of the masking and pointing) - that would be useful in tasks that we don't know the algorithm; but as it is, training a PGN for a specific task - requires us to have the oracle-algorithm to provide the step-by-step labels for that task. And if we have the oracle-algorithm, why do we need the neural network? We could encode all algorithms in history into neural networks and see how neural networks learn them - but would it be useful? Should we expect a series of papers that simulate all algorithms in history using GNNs using such a strong supervision? Does it advance at toward an (even slightly) better understanding of neural networks or a small step toward AGI? It is not surprising to me that GNN can learn a step-by-step execution of an oracle algorithm/data structure. It was more surprising if it could learn some of it "by itself" (i.e., with much less supervision, only input/output examples, and maybe some *minimal* additional guidance). * I am missing some discussion about the generality of this work, even without evaluating. What kinds of problems can PGNs address? What classes of algorithms/data structures are PGNs good in simulating? * Evaluation - the superiority of PGN-Ptrs over vanilla-PGN in the Disjoint-set task is bothering. It shows that a two-step distillation-like training works better: first, training to predict the new edges; then, train from scratch a standard GNN over the inferred edges. If this performs better, what advantage does joint training have? * Reproducibility - only the code for creating the data is available.

Correctness: The method seems to be correct.

Clarity: The paper is well written linguistically and mathematically. However, understanding the paper on the first read is difficult. It requires me to read the paper at least twice to understand it. I think that the authors could simplify writing and be more concrete earlier in the paper to help the reader (in the spirit of my sentence above: "Another (simpler?) way to think about it..."). Explaining the task (or a toy task) before diving into explaining the model would help the reader understand the scenario and the settings.

Relation to Prior Work: It seems that this paper is very related to: [42] ("Neural execution of graph algorithms", Velickovic et al., ICLR'2020) and to [50] ("Neural execution engines", Yan et al.). It is not completely clear what is the difference and novelty compared to these related work. It seemed that both [42] and [50] used a dynamic graph that learns its own adjacency ( [50] used a masked Transformer, but a masked Transformer is obviously equivalent to a GAT). Additionally, [50] also included the idea that each layer predicts a pointer. Is the difference only that this paper was trained on other algorithms that were not evaluated before, as stated in line 44?

Reproducibility: No

Additional Feedback: Questions to authors: * I am not sure about what exactly does the F1 metric measure? In a sequence of operations, does it measure the fraction of operations that were predicted correctly? What advantage does F1 have in this case over accuracy per timestep? And if the model is tested on a sequence of 30 ops, doesn't a mistake in the first op result in mistakes in the rest of the 29 ops, because the data structure is not consistent with the ground truth anymore? It would be interesting to see how the mistakes are distributed over "t", because of this strong dependence on previous time steps. * How does the model perform without pointers supervision? I.e., with only the final \hat(y) and \mu supervision signals, and allowing the model to use attention to as the pointers matrix \Pi? I.e., the attention scores (possibly using a threshold) are the next layer's \Pi, without direct supervision every layer? * How do the authors explain the superiority of PGN-Ptrs in the first task, and its inferiority in the second task (compared to PGN)? * Abstract & L42: "for improved model expressivity" - I'm not sure that saying that the expressivity is improved is correct. Theoretically, expressivity is the same. Maybe the authors should change that to "improved generalization ability"? * The authors argue that PGNs can learn "parallelisable data structures". I am not sure what exactly does this parallelism mean and how is it shown in the results/analysis. * L104: "computing entries of \Pi^t reduces to 1-NN queries in key/query space" - how is test time different than training time? And what does "1-NN" queries mean? * L197: does "||" denote concatenation? * Figure 2: what does "return 1// \hat{y}^t =1" mean? ============== Post-rebuttal comments ================== I have read the authors' response. I still think that this is a borderline paper, and I agree with the other reviewers that I would not like to see papers turning any classical algorithm into neural. I was not convinced that this is beneficial or practical. The paper does not discuss previous work ([42] and [50]) adequately enough, somewhat hiding their relatedness instead of stating their relateness and comparing their ideas directly. Specifically, the paper is called "pointer graph networks", presenting the "pointers" as the main novelty, while the idea of masked transformers with pointers for algorithmic problems was already presented in [50] and is very similar (even if for a slightly different goal). I increased my score to 6, because I think that the paper will be good enough for publication and interesting enough for the community. If this paper is accepted, I wish that the authors would have highlighted the related work ([42] and [50]) and explicitly discussed the novelty compared to them, instead of hiding the relatedness to them. I also wish that a discussion about the potential practical applications and their value would appear in the paper, and about the general class of problems that PGNs can be useful for - specifically in algorithms and data structures, and in other GNN problems in general.


Review 4

Summary and Contributions: This paper introduces Pointer Graph Network (PGN), a model that learns to infer a latent graph in order to simulate dynamic connectivity-based data structures such as disjoint-set unions and link/cut trees. At each time step, the model computes a sparse adjacency matrix of pointers (each nodes points to a single node, the one with the biggest attention coefficient, computed using a self-attention mechanism) and a mask that indicates the subset of nodes modified at the current time step. Those choices perfectly match the operations performed by the target data structures (DSU and LCT), representing an important theoretical inductive bias. The nodes representations are processed with a GNN that uses as adjacency matrix the pointer matrix, modified according to the computed mask. The supervision comes from the downstream task (the output obtained by running the desired algorithm on the input graph), but also from intermediate steps of the simulated algorithm. The experiments show that PGN is able to imitate, but also deviate from the target data structure, while also revealing out-of-distribution generalization.

Strengths: - Learning to simulate challenging algorithms on graph-data is an interesting problem and this work introduces a network able to execute the operations performed by two useful data structures: DSU and LCT. This result offers evidence that PGN-like modules can be used to solve more complex problems, aligned with these data structures such as connected components and minimum spanning trees. - The idea of creating a sparse adjacency matrix using the pointer mechanism is interesting and can be useful in other GNNs applications, beyond the domain of algorithms and data structures. - The experimental section is well done, showing superior results and also a stronger generalization for out-of-distribution input. (testing on graphs with a bigger number of nodes than those in the training set).

Weaknesses: - In Section 2, the authors mention that “To simplify the dataflow, we found it beneficial to symmetrise this matrix” (row 95). Is this actually a simplification of the method? In the target algorithm, it is crucial to follow the unidirectional recursion call (from child to parents). I think that by symmetrising the pointer matrix, the message passing benefits from a better exploration of the graph structure, leading to a more global representation for each node (not particularly necessary in this case). However, I don’t see why a node should send information both to children and to parents. (maybe by adding additional information on the edge, to differentiate between the two types of links child-parent or parent-child would make the process easier, while preserving the global view). - According to the paper, the processor applies only a single message-passing at each time step. Is it enough from the theoretical - algorithmic point of view? As long as the intermediate supervision urges the PGN to imitate the operations from the original algorithm, a number of iterations equal to the depth of the recursive call should be made (in order to reach the root of the current component). The height of each tree is probably very small, especially when using the compression trick. Is this the reason why a single step is enough? When increasing the number of nodes, eliminating the compression step or applying a more complicated algorithm, is it better to allow a variable number of iterations, computed as a function of size? - Running an additional ablative experiment, similar to the (Unrestricted) GNN but using the computed alpha coefficients as adjacency matrix (pi = alpha) would be great. It would indicate if a GAT architecture could learn sparser attention when it's necessary or it is mandatory to explicitly design that inside your network, using a pointer-network approach as in the current work.

Correctness: The idea presented in this paper is sound and all the components proposed in this work are well motivated.

Clarity: The paper is self-contained, both the method and experiments are clearly presented and easy to follow. Please see the “Additional feedback, comments…” box for some minor observations.

Relation to Prior Work: In my opinion, the paper covers well all the related work that I am aware of.

Reproducibility: Yes

Additional Feedback: Minor comments: - From the paper, it seems that randomize indexing is used to avoid ambiguity, since the nodes priorities are fixed in the initialization step (r_i). In this case, the amortised time complexity is close to the inverse Ackermann bound, according to [1], beeing much better than the one obtained with the coin-flipping method and reported in this work (O(n+f(1+log..)). - On row (166) using f to denote the number of calls to find() in the complexity equation is confusing since it could be read as “a function of…” - The motivation for PGN-Ptr is not clearly expressed in the paragraph. (rows 235-237), but only slightly mentioned later. - In Table 3 from the Supplementary, it would be useful to mention that miu is 1 when the node is *not* visited by the steps of the algorithm. Both from the writing and from the table it seems that it is the other way (miu = 0 when it is changed in the current step), but Eq. (7) contradicts this. [1] Ashish Goel, Sanjeev Khanna, Daniel H. Larkin, and Robert Endre Tarjan. Disjoint set union with randomized linking. ====================== UPDATE =============================== Thanks to the authors for their feedback. After reading the rebuttal, I keep my opinion regarding the acceptance of the article, since I find it valuable for the community. However, I agree with other reviewers' opinion that there are some elements that need to be better highlighted. First, since algorithmic reasoning is a pretty new field, discussion about the practical application of the current work would better emphasize the direction of the domain and motivates the importance of the proposed method for that community. Second, a clearer description of the differences between the proposed method and other previous neural algorithms ([42] and [50] as mentioned by other reviewers) in terms of contribution is necessary.

[Author Response · NeurIPS 2020]

Thank you to all reviewers for the very careful feedback. We respond here, and **we will update the paper accordingly**!

(**Experiments**) We evaluated additional models, providing further empirical evidence on the PGNs' benefits: (**SupGNN**)
baseline GNN whose latents have been supervised to be predictable of masks and pointers (without explicitly using this
information); (**PGN-Asym**) PGN without symmetrising pointers; (**PGN-MO**) PGN where only masks are supervised
(and pointers are not directly optimised); and (**DGM**) [19], a recent strong baseline method on latent graph inference.
Here we provide results on the **hardest** test set (LCT; $n = 100$; ops $= 150$); results on other sets mirror these findings.

| GNN | SupGNN | DGM | PGN-MO | PGN-Asym | PGN |
|---|---|---|---|---|---|
| $0.401_{\pm.123}$ | $0.541_{\pm.059}$ | $0.524_{\pm.104}$ | $0.558_{\pm.022}$ | $0.561_{\pm.035}$ | $\mathbf{0.616}_{\pm.009}$ |

(**R1 / R2 / R3**) Our work evaluates the extent to which existing neural networks are potent reasoning systems, and the
minimal ways (e.g. inductive biases / training regimes) to strengthen their reasoning capability. Hence our aim is not to
outperform classical algorithms, but make their concepts accessible to neural networks. PGNs enable reasoning over
edges not provided in the input, simplifying execution of any algorithm requiring a pointer-based data structure. PGNs
can find direct practical usage if, e.g., they are pre-trained on known algorithms and then deployed on tasks which may
require similar kinds of reasoning (with encoders/decoders "casting" the new problem into the PGN's latent space).

(**R1**) Inspired by your review, we scaled up the LCT test set to ($n = 200$, ops $= 300$) where the PGN ($0.636_{\pm.009}$)
catches up to Oracle-Ptr ($0.619_{\pm.043}$). We hope that this illustrates not only the robustness of PGNs to larger test sets,
but also provides a quantitative substantiation to our claim about ground-truth LCTs not having favourable diameters.

(**R1**) Our contribution is *both* in the inductive biases (e.g. pointers/masks) proposed and the provided supervision.
To strengthen this claim, we compare against **SupGNN** in experiments above. Supervising provides a clear boost in
performance, but is still insufficient to outperform PGNs, showing the importance of the architectural choices as well.

(**R2 / R1**) To strengthen our empirical contribution on learning latent graphs, we also compare against the recently
proposed **DGM** [19] loss function in the experiments above, finding PGNs extrapolate much better to larger test sets.

(**R2 / R4**) We added experiments on asymmetric pointers (**PGN-Asym**), with a clear drop in performance on the LCT
task. Corresponding to R4's comment, we find that while symmetrising the pointers is not strictly necessary, empirically
it is helpful because it makes the inferred graph slightly less sparse, enabling us to rectify for the case where the pointer
mechanism makes mistakes without sacrificing sparsity. In this sense, a more "global" view of each node is desirable.

(**R3**) Our work extends [42] by the addition of the **pointer / masking** inductive biases, allowing one to more efficiently
simulate algorithms for which the input graph does not correspond to the actual links used for reasoning, or cases where
an input graph is not provided at all. As we acknowledge, while conditional masking had been introduced in [50], here
we utilise masking to update node state rather than exclude it from the input—which can reduce information loss.

(**R3**) We now compare against **PGN-MO** in the experiments above, which supervises on **m**asks **o**nly and does not
directly optimise pointers. The results show that we can relax most of the supervision constraints and still outperform
baseline approaches (such as GNN and DGM), while for full returns an explicit pointer-based loss is required.

(**R3**) Please note that PGN-Ptrs requires *two passes*: first training a PGN, then re-using and fixing its pointers on a
second training pass. Allowing for dynamic adjusting of pointers works better on complex algorithms with less margin
for error, such as the LCT setup. If the PGN mechanism makes a mistake at any epoch, PGN training can meaningfully
recover from this while PGN-Ptrs don't have this flexibility. For DSU, path compression yields very shallow data
structures that are modellable in many equivalent ways, hence such errors are less punishing for PGN-Ptrs.

(**R3**) Thank you for remarking the correctness of our "expressivity" claim—we will amend as you suggested!

(**R3**) Various answers: $F_1$ measures performance across the (binary) query responses $y^{(t)}$, handling class imbalance. As
mistakes are measured on $y^{(t)}$, errors at step $t$ do not imply errors at step $t + 1$. By parallelisable, we imply that they
allow for using GPUs more effectively (e.g. the rollout vs. ground-truth in Fig. 4). 1-NN refers to 1-nearest neighbours,
for which efficient algorithms exist and we don't need to compute all $O(n^2)$ $\alpha_{ij}$ values. $\parallel$ is concatenation.

(**R4**) We agree that multiple GNN steps may be theoretically required to reconcile our work with teacher forcing the
data structure. We found it sufficient to consider only one-step here, but as tasks get more complex—especially when
compression is no longer applicable—dynamic number of steps (e.g. function of dataset size) is likely to be appropriate.

(**R4**) Thank you for suggesting the GAT experiment! While it seemed to compare favourably to GNN/DeepSets on the
(more compressible) DSU task, the unrestricted GAT model failed to get off the ground at all for the LCT task.

(**R4**) Thank you for the minor comments, which we will incorporate! Especially for pointing out the work of Goel *et*
*al.*, which allows us to report a tighter analysis on time complexity (as we effectively use randomised linking-by-index).

[Meta-Review · NeurIPS 2020]

We have strong and detailed reviews and there was considerable discussion. I agree with the comments in the reviews.